# Increased Radiation Sensitivity in Patients with Phelan-McDermid Syndrome

**DOI:** 10.3390/cells12050820

**Published:** 2023-03-06

**Authors:** Sarah Jesse, Lukas Kuhlmann, Laura S. Hildebrand, Henriette Magelssen, Martina Schmaus, Beate Timmermann, Stephanie Andres, Rainer Fietkau, Luitpold V. Distel

**Affiliations:** 1Department of Neurology, Ulm University, 89081 Ulm, Germany; 2Department of Radiation Biology, Erlangen University, 91054 Erlangen, Germany; 3Department of Oncology, Oslo University Hospital (The Norwegian Radium Hospital), 0424 Oslo, Norway; 4Department of Radiotherapy and Radiation Oncology, University Medical Center Hamburg-Eppendorf, 20246 Hamburg, Germany; 5Clinic for Particle Therapy at WPE, University Hospital Essen, 45147 Essen, Germany; 6Medicover München Ost MVZ, Humangenetik, 81667 Munich, Germany

**Keywords:** Phelan McDermid syndrome, SHANK3 deficiency, radiation sensitivity, chromosomal aberrations, atypical teratoid rhabdoid tumor

## Abstract

Phelan-McDermid syndrome is an inherited global developmental disorder commonly associated with autism spectrum disorder. Due to a significantly increased radiosensitivity, measured before the start of radiotherapy of a rhabdoid tumor in a child with Phelan-McDermid syndrome, the question arose whether other patients with this syndrome also have increased radiosensitivity. For this purpose, the radiation sensitivity of blood lymphocytes after irradiation with 2Gray was examined using the G0 three-color fluorescence in situ hybridization assay in a cohort of 20 patients with Phelan-McDermid syndrome from blood samples. The results were compared to healthy volunteers, breast cancer patients and rectal cancer patients. Independent of age and gender, all but two patients with Phelan-McDermid syndrome showed significantly increased radiosensitivity, with an average of 0.653 breaks per metaphase. These results correlated neither with the individual genetic findings nor with the individual clinical course, nor with the respective clinical severity of the disease. In our pilot study, we saw a significantly increased radiosensitivity in lymphocytes from patients with Phelan-McDermid syndrome, so pronounced that a dose reduction would be recommended if radiotherapy had to be performed. Ultimately, the question arises as to the interpretation of these data. There does not appear to be an increased risk of tumors in these patients, since tumors are rare overall. The question, therefore, arose as to whether our results could possibly be the basis for processes, such as aging/preaging, or, in this context, neurodegeneration. There are no data on this so far, but this issue should be pursued in further fundamentally based studies in order to better understand the pathophysiology of the syndrome.

## 1. Introduction

Autism spectrum disorders (ASDs) are characterized by deficits in social interaction, limitations in communication and repetitive, stereotyped behaviors and special interests [1]. A prevalence of 0.9–1.1% for autism spectrum disorders can be assumed [2]. Based on twin and family studies, the heritability of autism spectrum disorders is around 40% [3]. Copy-number variations or de novo mutations in the *SHANK* genes occur in around 1% of all patients with ASD [4], so that these are among the most common ASD-associated genes, with *SHANK3* accounting for the majority at 0.73% [4,5]. SHANK3 stands for SH3 domain and ankyrin repeat-containing protein and is located on the long arm of chromosome 22, encoding structural proteins of the postsynapse of excitatory neurons [6]. Deletions or pathogenic variants of *SHANK3* lead to the clinical picture of Phelan-McDermid syndrome (PMS, OMIM#606232), which can be considered representative of syndromic autism as a monogenetic form of ASD [5]. PMS is characterized by global developmental delay with motor hypotonia, expressive and receptive language development delay, cognitive impairment and other neuropsychiatric comorbidities, such as mood disorders, epilepsy, regression, schizophrenia and autism spectrum disorder [7]. The latter is of clinical and developmental relevance in up to 70% of these patients [8]. The clinical variability is wide, a circumstance that has not yet been conclusively explained pathophysiologically, although there are first indications of epigenetic changes on chromosome 22 [9,10].

The genetic changes underlying the syndrome also vary greatly and include point mutations, deletions, inversions and translocations affecting *SHANK3* on chromosome 22q13.3 [10]. By now, patients with genetic aberrations in the chromosomal region 22q13 without *SHANK3* involvement are also known to show a similar phenotype so that the nomenclature of the syndrome was recently adjusted to PMS-*SHANK3* related or PMS-*SHANK3* unrelated [11]. The *SHANK3* gene product has no known linkage to DNA repair, cell cycle control or cell death control. It is thought that perturbations in these functions may lead to increased radiation sensitivity. A gene located near *SHANK3* and associated with radiation sensitivity is the *PPP6R2* gene, which encodes protein phosphatase 6 [12]. It is also located on 22q13.3, 300kb upstream of *SHANK3*. Another gene that is certainly associated with radiation sensitivity is *XRCC6,* positioned at 22q13.2, encoding the KU70 protein. This protein is important for the initiation of DNA double-strand break repair [13]. Even more distant (22q11.23) is the *SMARCB1* gene, in which pathogenic germline variants are associated with the occurrence of atypical teratoid/rhabdoid tumors that are more common in PMS patients [14].

Radiation sensitivity refers to both stochastic and deterministic risk [15]. On the one hand, this means an increased risk of cancer susceptibility even at low doses of ionizing radiation (IR), and at higher doses, as in radiotherapy, an increased risk of adverse therapeutic effects. Radiation sensitivity already varies considerably in healthy individuals but can be greatly increased by a factor of 2–4 in genetic syndromes, such as ataxia-telangiectasia or Nijmegen breakage syndrome [16]. There are many other autosomal recessive disorders in which only limited increased radiation sensitivity occurs. These are, e.g., Fanconi anemia, Ligase IV, Bloom syndrome, Werner syndrome, Xeroderma pigmentosum and Cockayne syndrome [17]. Lymphocytes and the analysis of chromosomal aberrations after ex vivo irradiation are particularly suitable for testing individual radiation sensitivity [18]. The main reason why lymphocytes are particularly suitable is that they are in the G0 cell cycle phase and are, therefore, uniformly sensitive to radiation. In addition, they then pass through the entire cell cycle control system before the metaphases are then analyzed to determine radiation sensitivity.

In an affected individual with Phelan-McDermid syndrome suffering from an atypical teratoid/rhabdoid tumor (AT/RT) WHO grade 4, individual radiation sensitivity was analyzed prior to the start of radiation therapy and was significantly increased compared with the control cohort. Therefore, this pilot study was undertaken with the question of whether increased radiation sensitivity also exists in other patients with Phelan-McDermid syndrome.

## 2. Materials and Methods

The blood of 20 PMS patients was compared with that of 591 patients in a control cohort and used to study radiation sensitivity. Thus, 9 of the PMS patients were female and 11 were male. The mean age was 9.8 years with a range of 3.5 years to 31.8 years. The patients varied widely with respect to disease burden, with impairments in receptive and expressive communication, motor skills and cognition (Table 1). Eight of them suffered from autism and five from epilepsy. The severity of disease was scored according to the clinical categorization in Table 1. High values represent a high clinical burden of disease. The highest possible score for the most severe clinical impairment is 14. PMS patients were compared with 218 healthy subjects, 226 patients with rectal cancer and 147 patients with breast cancer. The healthy subjects were 57.3% female and had a mean age of 50.4 years, the rectal cancer cohort was 28.3% female/63.2 years, and the breast cancer cohort was 100% female/57.3 years. Due to the young age of PMS patients, subgroups were formed with healthy subjects younger than 30 years, resulting in 51 subjects, of whom 58.8% were female and the average age was 25.6 years. Cancer patients are older, so a higher age of 45 years was used to form a young group. Patients with rectal cancer were 50% female and had a mean age of 35.5 years; breast cancer patients were 100% female and had a mean age of 39.1 years. All blood samples were taken before the start of radiotherapy. Some of the data have already been published [19,20]. All patients and healthy individuals gave their written informed consent for the scientific processing of their material and data. The ethics committee of the University Hospital Erlangen approved the study, including the use of the individual patient data. Approval of the Ethics Committee 21_19 B.

The G0 three-color fluorescence in situ hybridization (FISH) assay was used to determine the radiation sensitivity of patients using blood lymphocytes. For this purpose, 9 mL of blood was drawn from each patient and half of the blood was irradiated with 2 Gray (Gy) of a 6 MeV linear accelerator (Versa HD, Elekta, Stockholm, Sweden). The other half remained unirradiated to determine the background of chromosomal aberrations. Lymphocytes were stimulated by phytohemagglutinin to undergo the cell cycle and, 47 h later, cells were arrested in metaphase by colcemid. Lymphocytes were prepared on slides and chromosomes were hybridized by probes for chromosomes #1, #2 and #4. Images were acquired via a Zeiss fluorescence microscope (Axioplan Z2, Zeiss, Göttingen, Germany) and an automatic metaphase finder (Metasystems, Metapher 4, Altlussheim, Germany). Chromosomal aberrations in the three chromosomes studied were analyzed using image analysis software (Biomas, Erlangen, Germany). The number of possible DNA breaks representing the different chromosomal aberrations was scored according to Savage and Simpson [21]. Acentrics were counted as one break, dicentrics, translocations and rings as two breaks and insertions as three breaks. Complex aberrations were scored according to how many DNA double-strand breaks would have been necessary for their formation. An average value of breaks per metaphase (B/M) was derived from at least 200 metaphases from unirradiated and 150 metaphases from 2 Gy irradiated samples [20].

The SPSS Statistics 28 program (IBM, Armonk, NY, USA) was used for analyses and statistical work. Statistical significance was calculated with the *t* test for cohorts n > 30 and with the Mann–Whitney U test for groups with *n* ≤ 30. Tests were always performed two-sided. Charts were generated using Excel (Microsoft Corporation, Redmond, WA, USA) and Prism (GraphPad Software, San Diego, CA, USA).

## 3. Results

### 3.1. A Case of a Phelan-McDermid Syndrome Patient with an Atypical Teratoid/Rhabdoid Tumor

A 6-year-and-8-month-old boy was diagnosed with an atypical teratoid/rhabdoid tumor (AT/RT), WHO grade 4, in the fourth ventricle/aqueduct in 2018. The boy was known to have Phelan-McDermid syndrome (PMS). A 3.2Mb deletion was diagnosed in the terminal end of chromosome 22 in region 22q13. The boy’s symptoms included unspecified intellectual disability, impaired speech and sleep problems. A change in health status was characterized by stiffness and reclusiveness, suspected epilepsy and focal activity on electroencephalogram. The tumor was diagnosed by MRI. It was resected, with no residual tumor visible. Chemotherapy was administered due to EU-RHAB V5 2016 protocol, consisting of three cycles of doxorubicin (Dox), ICE (Ifosfamide, Carboplatinum, Etoposide) and VCA (Vincristine, Cyclophosphamide, Actinomycin) and an intraventricular injection of methotrexate for 5 months (Figure 1A). No evidence of tumor was found on MRI examinations performed at regular intervals. Proton radiotherapy was originally planned according to the EURHAB protocol with 30 × 1.8 Gy single fraction dose up to a total dose of 54 Gy (Figure 1B,C). The therapy was started immediately due to the aggressiveness of the tumor. In parallel, radiation sensitivity testing was performed by irradiation of whole blood in G0 ex vivo and chromosome aberration analysis by 3-color fluorescence in situ hybridization (Figure 1D–G). A significantly increased rate of chromosomal aberrations of 0.74 breaks per mitosis was found. Due to these results, it was decided to decrease both the fraction dose and the total dose. After 11 fractions had already been applied with 1.8 Gy per fraction, the single dose was reduced to 1.4 Gy per fraction and given for a further 8 fractions. Therefore, the total dose was reduced to 31 Gy. Since then, MRIs have been performed on a regular basis. Until the last MRI in June 2022, no progression of disease nor any late effects occurred. In July, the mother reported that the boy was doing well, with delayed development due to PMS.

### 3.2. Individual Radiation Sensitivity of 20 Patients with Phelan-McDermid Syndrome

Because of the increased radiation sensitivity of the index patient, we were interested to know if other individuals with PMS were similarly more radiosensitive. Therefore, we studied the radiation sensitivity of 11 males and 9 females with PMS with an average age of 9.8 years (Table 1). The PMS cohort with an average radiation sensitivity of 0.653 B/M had a clearly increased radiation sensitivity compared to cohorts consisting of healthy individuals with an average radiation sensitivity of 0.417 B/M, patients suffering from rectal cancer with an average radiation sensitivity of 0.434 B/M and breast cancer with an average radiation sensitivity of 0.489 B/M. Because of the young age of the PMS cohort, additional subgroups of these cohorts were formed, consisting of healthy individuals younger than 30 years (mean radiation sensitivity 0.377 B/M), rectal cancer patients (0.476 B/M) and breast cancer patients (0.562 B/M) younger than 45 years (Table 2).

Background aberrations in lymphocytes among the PMS patients (0.0131 B/M) were very low, with the exception of one individual (Figure 2A). The PMS background aberrations were also clearly lower compared to the young cohorts with cancer (healthy 0.0182; rectal cancer 0.0549; breast cancer 0.0996 B/M *p* < 0.017). However, there was no difference to the healthy group (*p* = 0.213) (Figure 2B). The chromosomal aberrations induced by 2 Gy after background withdrawal displayed for the whole cohorts (*p* < 0.001), as well as for the young cohorts (*p* < 0.041), that the PMS cohort had a strongly increased radiation sensitivity, with an average B/M of 0.653. Only 2 of the PMS patients had no increased radiation sensitivity by our definition, and in 17 of the 20, the radiation sensitivity was increased to a value where dose reduction would be recommended in the event of radiotherapy (Figure 2C,D, Table 3). This is particularly noteworthy, as nearly all PMS individuals had increased radiation sensitivity. This is in contrast to the other cohorts, where only a few individuals had an increased radiation sensitivity, whereas the majority still had average radiation sensitivity.

### 3.3. Mutation Type and Radiation Sensitivity

Next, we were interested in whether the locations and the type of the genetic aberrations are associated with the increased radiation sensitivity. There were six frameshift mutations, one nonsense mutation and one intragenic deletion within the SHANK3 gene (0.634 B/M) (Figure 3). All variants were located in exon 20 of SHANK3 and there was no difference in radiation sensitivity in these cases.

Next, we studied the association of increased radiation sensitivity in the seven subjects carrying a large deletion, including the SHANK3 gene, four subjects with relatively small deletions, not including the SHANK3 gene, and one subject with a ring chromosome. These patients had a radiation sensitivity of 0.666 B/M (Figure 4). We did not find any difference in the radiation sensitivity of the subjects with a deletion compared to subjects with pathogenic variants in the SHANK3 gene. The ring chromosome had a very high radiation sensitivity (0.8 B/M), while the relatively small (mean 128 kb) terminal deletions not encompassing the SHANK3 gene had a slightly lower mean (0.619 B/M) compared to the larger (mean 4.2Mb) deletions encompassing the SHANK3 gene with a higher mean (0.668 B/M) (*p* = 0.257) (Figure 4 and Figure 5A).

Next, we studied whether a high burden of symptoms was related to the increase in radiation sensitivity. The individuals had markedly different limitations in terms of impaired receptive and expressive communication, motor skills, cognitive, autistic disorders, epilepsy and gastrointestinal disorders (Table 1). We formed a score with eight items (Table 1), with a maximum of 14 points representing the highest burden. The median score was 6 points, with a range from 0 to 12 points. Patients with a high disease burden above the median had an average radiation sensitivity of 0.686 B/M compared with patients with a lower disease burden of 0.620 B/M, but without a clear difference (*p* = 0.290) (Figure 5B). Additionally, we were interested in whether the genetic variants were related to the disease burden score we used (Figure 5C). We could only find a tendency towards a higher disease burden in the group that was not related to SHANK3 (*p* = 0.170), with the limitation that the groups are very small. The patient with the ring chromosome had the highest disease burden score, which also had significantly increased radiation sensitivity.

## 4. Discussion

The PMS cohort had massively increased radiation sensitivity compared to the three control cohorts. A peculiarity compared to the studied cancer cohorts is that almost all individuals with PMS are affected by this clearly increased radiation sensitivity. Even though the studied group of 20 individuals was not very large, it must be assumed that a large proportion of patients with PMS have increased radiation sensitivity. To our knowledge, this is the first study of PMS patients for individual radiation sensitivity. We found no difference in patients, regardless of whether the syndrome was classified as *SHANK3*-related or *SHANK3* non-related. It is remarkable that PMS patients with *SHANK3* non-relation are only slightly less sensitive to radiation than patients with *SHANK3* relation. This study was prompted by a patient with an atypical teratoid/rhabdoid tumor whose individual radiation sensitivity was significantly elevated at 0.74 B/M. The proton therapy dose was, therefore, reduced from a total dose of 54 Gy to 31 Gy to avoid late effects in normal tissue [22,23]. Four years after radiotherapy, the patient was still free of disease, despite the radiation dose being significantly lowered. This is particularly noteworthy, as this tumor is highly aggressive, often leading to tumor recurrence or progression [24]. The successful treatment with relatively low doses could indicate that a reduction in the dose was adequate. So far, no late adverse events have occurred. Contrast this with the case of a 12-month-old girl with PMS who developed AT/RT. She was treated with surgery, induction chemotherapy and moderate radiotherapy with 1.64 Gy single doses up to a total dose of 45.9 Gy, followed by three cycles of chemotherapy. Over the next year and a half, MRI revealed diffuse volume loss in both the cerebral hemispheres and brainstem, myelomalacia in the cervical spinal cord and multiple foci of diffusion restriction, consistent with progression of brainstem necrosis. Symptoms included acute right-sided paralysis, global hypotonia, verbalization arrest and development of acute encephalopathy and dysautonomia, with intermittent bradycardia, hypotension and hypopnea [25]. This indicates a PMS patient who developed a very severe side effect from a moderate dose of IR. This is in line with our observation that patients with PMS have significantly increased radiation sensitivity.

The radiation sensitivity of the PMS cohort was compared with that of healthy subjects and cancer patients and with the same cohorts but limited to young subjects. The reason for this is that radiation sensitivity can change with age [19,20]. There are two opposing effects here. First, the DNA damage response becomes weaker with age, leading to a slight increase in radiation sensitivity with age [26,27]. The opposite is true in cancer patients, where younger subjects often suffer from genetic alterations that lead to early onset of cancer and increased radiation sensitivity at the same time [20,28]. The young patients in our breast cancer cohort were significantly more radiosensitive compared to the overall cohort, whereas the PMS patients were still significantly more radiosensitive on average.

We equate, here, the chromosomal aberrations induced by 2 Gy IR as a measure of radiation sensitivity. It has been shown for a long time that chromosomal aberrations are best suited for this purpose. We used a three-color FISH approach because the three largest chromosomes can be used to check about 22% of the total DNA [29,30]. In contrast to commonly used conventional cytogenetics, in addition to dicentric and acentric aberrations, FISH can effectively find aberrations, such as insertions and translocations [29,30]. FISH with more than 3 chromosomes and up to 24 chromosomes is increasingly difficult to score and offers no advantage in reliability of prediction from radiation sensitivity over the three-color FISH [31]. In the studied PMS individuals, the average radiation sensitivity was increased by 56.6% compared to healthy individuals and by 73.2% compared to the young healthy cohort. If one does not want to make the connection with radiation sensitivity directly, it is nevertheless the case that PMS individuals have a 56% to 73% increased mutation rate in the chromosomes. Nevertheless, a question arises about the pathophysiological cause of the detected increased radiation sensitivity. It does not appear to be only dependent on *SHANK3,* as no association of genotype with radiation sensitivity was seen, keeping in mind that the groups studied are relatively small. Epigenetic changes may also play a causal role here, which is certainly interesting to follow up.

Another question according to the clinical relevance of the data arises. In patients with Phelan-McDermid Syndrome, the development of cancers seems not markedly increased compared to other radiation sensitive diseases, such as Li Fraumeni syndrome, Fanconi anemia, Ataxia telangiectasia and Nijmegen breakage syndrome. Tumors that occur in PMS typically include rhabdoid tumors, as in our index patient [14] and neurofibromas involving the *NF2* gene in patients with PMS and ring chromosome [32]. Atypical teratoid/rhabdoid tumors arise from pathogenic mutations in *SMARCB1* [33] that are also located on chromosome 22q (22q11.2) (Figure 4). The atypical teratoid/rhabdoid tumor is frequently described in patients with PMS [14], and there may be a link between PMS and downregulation of genes on chromosome 22 through epigenetic changes [34]. Based on the current data in the literature on the incidence of tumors in Phelan-McDermid syndrome, the increased radiation sensitivity does not appear to be associated with an increased tumor tendency. It remains to be debated whether the increased radiation sensitivity may lead to premature aging of the disease, starting early in development, as Braak described for Alzheimer’s disease, since we already detect a significantly increased radiation sensitivity in young children in our cohort [35]. Indications for this could be the description that some patients with PMS develop a dementia syndrome in the course of their lives [36,37]. Systematic studies on this are still lacking but would be exciting.

Another explanation could be that selective radiation sensitivity occurs in PMS, which is relatively specifically susceptible to ionizing radiation. Other predisposition syndromes, such as *BRCA1-/BRCA2*-related tumor predisposition syndrome, Fanconi anemia and Li Fraumeni syndromes, are sensitive to both IR and chemicals, as they interfere with different repair pathways that are also responsible for chemicals. In contrast, PMS could interfere with the repair of lesions specifically caused by the relatively rare environmental IR. In addition, a specifically increased incidence of IR-induced chromosomal aberrations could also be caused by PMS, so that chromosomal aberrations are clustered and other mutations are not correspondingly frequent. This could then mean that the radiation sensitivity is not quite as high as here from the measurements. It would, therefore, be important to study radiation sensitivity in PMS using other methods and on other cells such as fibroblasts.

We suspected that larger deletions could lead to both a higher disease burden and increased radiation sensitivity [38]. However, there is only a slight tendency for this association. To date, several studies have found correlations between deletion size and phenotype [39,40,41,42,43] but results are conflicting. Two recent large cohort studies also addressed genotype and phenotype correlations [44,45]. The latter is the largest study to date, in terms of the number of individuals with PMS, with 210 participants (including 21 individuals with a *SHANK3* variant) from Spain. The authors observed that there was a positive correlation of deletion size with brain MRI findings, ear abnormalities and toe syndactyly, while abnormal behavior correlated negatively. Interestingly, individuals with large deletions were more likely to have macrocephaly, while small deletions were associated with microcephaly. A French study [44] subdivided 22q13.3 deletions into two classes: Class I included small deletions containing *SHANK3,* to which the authors added those with *SHANK3* pathogenic variants; Class II contained all other (larger) deletions located on chromosome 22q. Class I individuals had a less pronounced delay in development and higher cognitive abilities but were more prone to regression of skills and psychiatric disorders. Class II individuals more often had abnormalities of the kidneys, eyes and spine. Ataxia and muscular hypotonia were also more common.

## 5. Conclusions

Although our pilot study revealed very interesting results with increased radiation sensitivity in patients with PMS, the results should be interpreted with caution due to the small number of patients and the arbitrary determination of our clinical score. It would be interesting to reproduce the results in a larger group of patients with PMS.

It can certainly be concluded that recommendations should be given to help analyze radiation sensitivity in patients with PMS when planning any radiation therapy. In addition, the results of this pilot study suggest that a reduction in the radiation dose must be considered in PMS and discussed with affected families.

## Figures and Tables

**Figure 1 cells-12-00820-f001:**
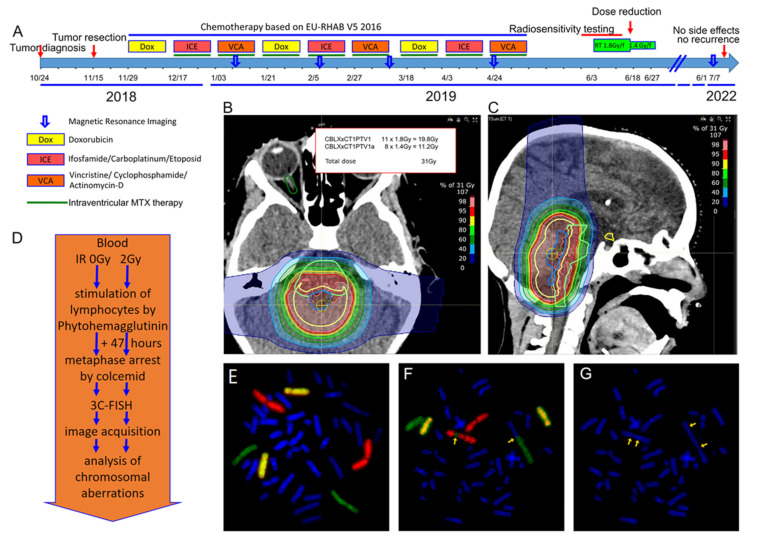
Case of a 6-year-and-8-month-old boy and radiation sensitivity testing. (**A**) Timeline since tumor diagnosis. (**B**) Proton therapy dose plan for the atypical teratoid/rhabdoid tumor (**B**) in a transverse plane and (**C**) a sagittal plane. (**D**) Timeline of radiation sensitivity assay. (**E**) A metaphase with the three stained chromosome pairs without aberrations. Red-stained chromosome #1, green-stained chromosome #2 and yellow-stained chromosome #4. (**F**) A metaphase with an insertion of a fragment of chromosome #2 into chromosome #1 and a dicentric chromosome consisting of chromosome #2 and an unstained chromosome. The yellow arrows point to the chromosomal aberrations. (**G**) The same image with DAPI staining shows that both damaged chromosomes are dicentric. The yellow arrows point to the centromeres.

**Figure 2 cells-12-00820-f002:**
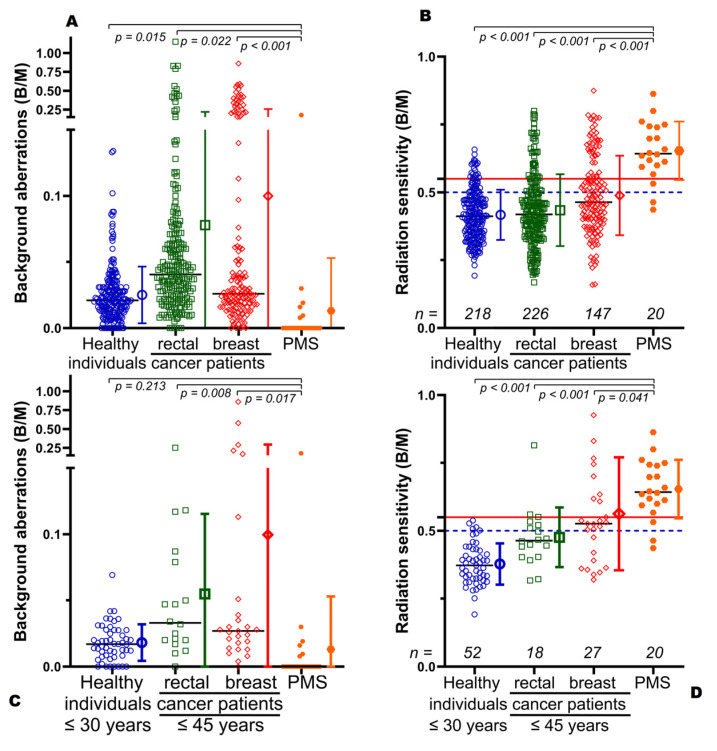
Radiation sensitivity testing in 20 subjects with Phelan-McDermid syndrome. Chromosomal aberrations in ex vivo irradiated lymphocytes from 20 individuals with Phelan-McDermid syndrome compared with a cohort of healthy individuals, patients with rectal cancer and breast cancer. (**A**) Background aberrations of all ages from 9 to 91 years and (**B**) of young healthy individuals younger than or equal to 30 years and cancer patients younger than or equal to 45 years. Chromosomal aberrations after ex vivo irradiation with 2 Gy IR and subtraction of background aberrations give (**C**) radiation sensitivity of the whole cohort or (**D**) young individuals. B/M = Breaks per metaphase. The dashed blue line indicates individuals with increased radiation sensitivity and the solid red line indicates individuals with significantly increased radiation sensitivity.

**Figure 3 cells-12-00820-f003:**
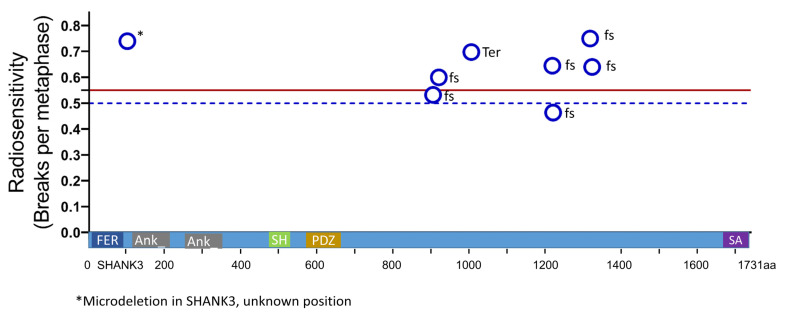
Location of pathogenic variants in the *SHANK3* gene in eight patients with PMS and associated radiation sensitivity expressed as breaks per metaphase. The location of the variants in the SHANK3 gene of eight individuals with PMS is indicated by blue rings. The height indicates the associated radiation sensitivity (fs = frameshift mutation and Ter = termination mutation). The mutation marked by an asterisk is a microdeletion in the SHANK3 gene with unknown location. The dashed blue line indicates individuals with increased radiation sensitivity and the solid red line indicates individuals with significantly increased radiation sensitivity.

**Figure 4 cells-12-00820-f004:**
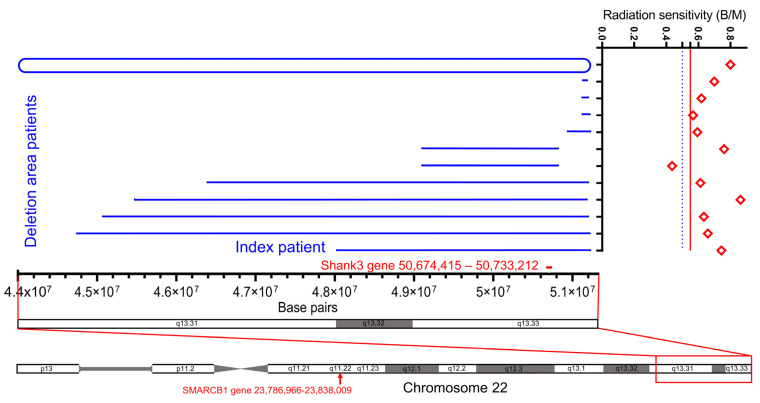
Deletion regions of chromosome 22 in 12 PMS patients and their radiation sensitivity expressed as breaks per metaphase. The entire chromosome and a subsection (red box) of chromosome 22 are indicated. The location of the SHANK3 gene is indicated as a short red line. The blue lines indicate the length of deletions in 11 individuals. The upper closed line indicates an individual with a ring chromosome. Radiation sensitivity of individuals is indicated in the right column. The dashed blue line indicates individuals with increased radiation sensitivity, and the solid red line indicates individuals with significantly increased radiation sensitivity. Loss of function of the SMARCB1 gene (red arrow) and its product, the tumor suppressor protein INI1, is associated with tumor incidence and, in particular, with atypical teratoid/rhabdoid tumors.

**Figure 5 cells-12-00820-f005:**
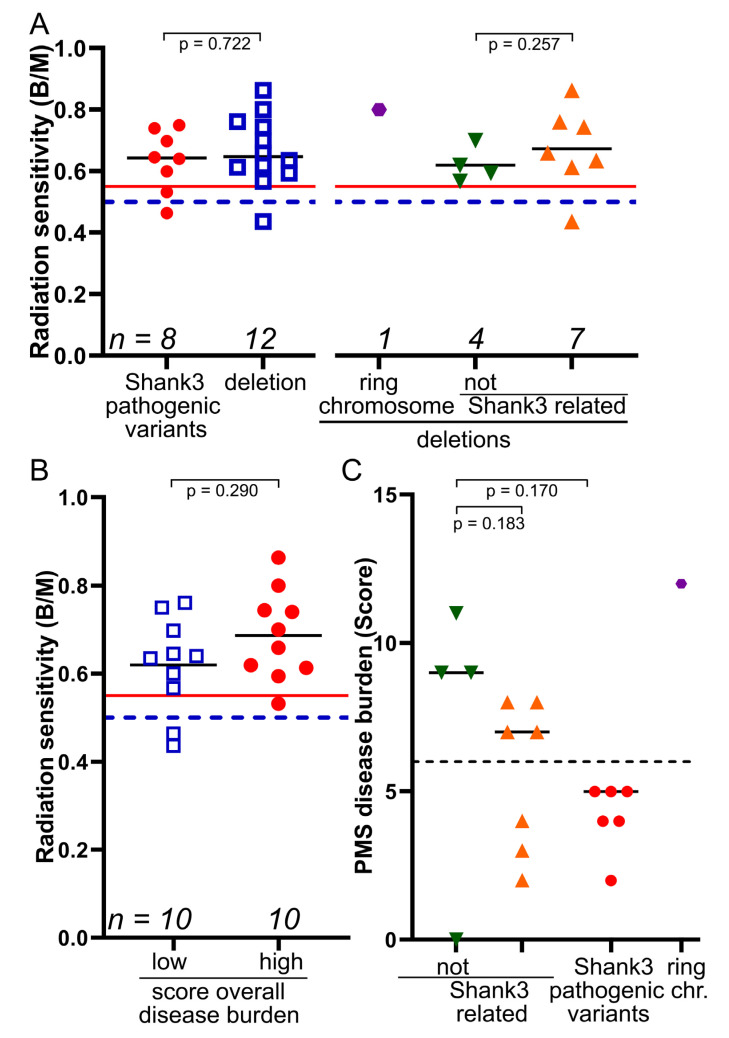
Association of pathogenic variant locations and burden of disease with radiation sensitivity. (**A**) Radiation sensitivity associated with PMS pathogenic variants within the gene compared with deletions in the terminal region of chromosome 22q, including the ring chromosome and ring chromosome compared with deletions, not including *SHANK3.* (**B**) A score of 14 points for the most severe disease burden was formed from all clinical characteristics. The low group had ≤6.0 points and the high group had > 6.0 points. (**C**) Pathogenic variants within the *SHANK3* gene compared with deletions in the terminal region of chromosome 22q (unrelated or associated with the *SHANK3* gene) and their association with a PMS disease burden score. B/M = Breaks per metaphase. The dashed blue line indicates individuals with increased radiation sensitivity and the solid red line indicates individuals with significantly increased radiation sensitivity. The dashed black line indicates the median PMS disease burden score. chr. = chromosome.

**Table 1 cells-12-00820-t001:** Clinical characteristics and symptoms of 20 patients with Phelan-McDermid syndrome. A disease burden score was built from these symptoms to assess disease burden. The possible scores for the various clinical symptoms are given, with a high number of points representing the most severe disease burden in each case. An additional point could be given for further symptoms.

Variable	Phelan McDermid Syndrome *n* = 20 (%)	Score (pnts)
sex (male; female)	11 (55%); 9 (45%)	-
age (range)	9.8 (3.5–31.8)	-
impaired communication receptive (minimal; light; medium; heavy)	7 (35%); 5 (25%); 5 (25%); 3 (15%)	3
impaired communication expressive (minimal; light; medium; heavy)	1 (5%); 9 (45%); 7 (35%); 3 (15%)	3
motor skills (normal; mild hypotonia; moderate–severe hypotonia; fine motor disturbance)	1 (5%); 13 (65%); 2 (10%); 4 (20%)	3
cognition (not tested; below average intelligence level)	14 (70%); 6 (30%)	1
autism (no; yes; no data)	11 (55%); 8 (40%); 1 (5%)	1
epilepsy (no; yes; no data)	14 (70%); 5 (25%); 1 (5%)	1
complaints of the gastrointestinal tract (no; yes; no data)	17 (85%); 2 (10%); 1 (5%)	1

**Table 2 cells-12-00820-t002:** Clinical characteristics of control cohorts.

Cohort (n)	Variable	Total Cohort	Subgroup “Young“ Patients
Healthy individuals *n* = 218	sex (male/female)	93 (42.7%)/125 (57.3%)	21 (41.2%)/30 (58.8%)
	age (range)	50.4 (9–81)	25.6 (9–30)
Rectal cancer *n* = 226	sex (male/female)	162 (71.7%)/64 (28.3%)	9 (50%)/9 (50%)
	age (range)	63.2 (23–87)	35.5 (23–45)
	Primary tumor (T1/T2/T3/T4)	7 (3%)/28 (12.2%)/146 (64.8%)/45 (20.1%)	1 (3%)/3 (15.2%)/12 (66.7%)/3 (15.2%)
	Regional lymph nodes (N0/N1)	86 (38.2%)/140 (0%)	6 (33.3%)/12 (66.7%)
	Distant metastasis (M0/M1)	185 (81.8%)/41 (18.2%)	15 (81.8%)/3 (18.2%)
Breast cancer *n* = 147	sex (female)	147 (100%)	27 (100%)
	age (range)	57.3 (28–91)	39.1 (28–45)

**Table 3 cells-12-00820-t003:** Radiation sensitivity in the different cohorts.

	Mean Radiation Sensitivity (B/M)		Increase Compared to PMS (%)	
	All	Young	All	Young
PMS	0.653	0.653	-	-
Healthy	0.417	0.377	56.6	73.2
Rectal cancer	0.434	0.476	50.5	37.2
Breast cancer	0.489	0.562	33.5	16.2

## Data Availability

All data are available in the main text. Further data can be obtained from the corresponding author on request.

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
