# Peer review of "Increased Radiation Sensitivity in Patients with Phelan-McDermid Syndrome"

_cells, 2023, doi:10.3390/cells12050820_

Round 1

Reviewer 1 Report

I enjoyed reading the article and the information in the introduction about PMS is correct.  I am not a cancer expert at all and still found it very interesting.  That said I have several comments:

1) Abbreviations like Gy and DSBs should not be used without prior definition of the term.

2) Why were chromosomes 1,2,4 only looked at for breaks?  This should be explained with references.

3) Remove the paragraph directly under section 3 for results.

4) Typical genetic terminology for a deletion- line 140- is 3.2 Mb

5) line 142- intellectual disability rather than mental retardation

6) line 177- please include some form of IQ range even if you use severe ID, etc

7) Figure 1A- Need further clarification of patient status besides saying "patient is OK"

8) lines 241 and 242- sentence needs to be rewritten- what does encompassing or not mean?

9) Figure 3- I assume the intragenic deletion was detected with a microarray which should be able to give size and location of the deletion if there was adequate probe coverage 

10) Line 259- a deletion of 536,957 kb equals 536 Mb which roughly one sixth of the entire genome and much larger than the entire 22nd chromosome!  The typical deletion in PMS is 3-7 Mb with the largest around 13 Mb.

11) Figure 5A- you studied 20 patients but show 8 gene variants (please use pathogenic variants rather than mutation), 12 deletions and 1 ring- that adds up to 21.  Why are your numbers inconsistent?  Please better define "not SHANK3 related- what does this mean and where were these patients previously introduced

12) Sentences from 303 to 307 are not clearly written

13) line 336-337 should read which will require further investigation

14) space between ring and chromosome at line 343

12) 

Reviewer 2 Report

This scientific paper is a very actual topic. Radiossensitivity to ionizing radiation related to hereditary syndromes is very important being the information very important not only for the therapeutics but also for the diagnostics approach.

Concerning the follow topics the authors can improve the paper according.

Abstract – In my opinion the abstract should be improved concerning the results obtained should be more represented in the abstract.

Introduction- In this section could be important to give more information about the syndrome in general and concerning also the radiation I think the function of genes could be more contextualized in terms of ionizing radiation effects. One of the last paragraphs ..." This because all lymphocytes...." should be in the methods section 

Discussion: Concerning the results obtained and have in consideration that the age was a characteristic that influence ionizing radiation effects I think this topic could be more developed in this section.  

In my opinion this paper should be submitted to huge alterations before be accepted. My expertise is related with to radiobiology in my point of view are several aspects that could be altered like the genetic information.
